# Decoupled Context Processing for Context Augmented Language Modeling

**Zonglin Li** [*]
Google Research, New York
lizonglin@google.com

**Ruiqi Guo** [*]
Google Research, New York
guorq@google.com

**Sanjiv Kumar**
Google Research, New York
sanjivk@google.com

## Abstract

Language models can be augmented with a context retriever to incorporate knowledge from large external databases. By leveraging retrieved context, the neural network does not have to memorize the massive amount of world knowledge within its internal parameters, leading to better parameter efficiency, interpretability and modularity. In this paper we examined a simple yet effective architecture for incorporating external context into language models based on decoupled `Encoder-Decoder` architecture. We showed that such a simple architecture achieves competitive results on auto-regressive language modeling and open domain question answering tasks. We also analyzed the behavior of the proposed model which performs grounded context transfer. Finally we discussed the computational implications of such retrieval augmented models.

## 1   Introduction

Transformers have proven to be powerful language models that capture an impressive amount of world knowledge in its internal parameters and generalize to a variety of downstream tasks [39, 31]. Recently, there has been a lot of success in improving language model quality by increasing the number of parameters in transformers, often on the order of hundreds of billion [10, 30, 38, 6]. However, the scaling of model size also contributes to the exponential rise of the computation costs, both in terms of the number of accelerators needed and energy consumption [29].

To overcome the exponential increase in the number of parameters, one natural idea is to utilize information retrieved from an external source such as a massive external database, therefore freeing the neural network from having to memorize world knowledge. To this end, researchers proposed multiple context augmented language model architectures [21, 15, 4, 44, 26]. Such architecture typically has two components: a retriever that embeds the input sequence and retrieves relevant context from external source through vector similarity search; a neural network that integrates both the input and retrieved external context into the prediction of target sequence, formally:

$$P(\mathbf{y}|\mathbf{x}, \mathbf{C} = Retrieve(\mathbf{x}, \mathcal{D}); \theta) \geq P(\mathbf{y}|\mathbf{x}, \theta') \tag{1}$$

Here, $\mathbf{C} = \{\mathbf{c}\}$ is a set of context retrieved from the external database $\mathcal{D}$. $\theta'$ is a self-contained language model which predicts target sequence $\mathbf{y}$ based solely on the input $x$ whereas $\theta$ corresponds to the context augmented language model which incorporates both the input $\mathbf{x}$ and the retrieved context $\mathbf{C}$.

One of the challenges for such context augmented language model is the computational cost of context retrieval and incorporation, especially when multiple pieces of context is present or the context sequence is long. In this paper, we propose a computationally efficient architecture for

---

[*]Equal contribution

36th Conference on Neural Information Processing Systems (NeurIPS 2022).

incorporating context based on vanilla `Encoder-Decoder`, which decouples the encoding of context and the prediction of target sequence. We show that the model with such a simple architecture is competitive when compared with customized mechanisms such as Chunked-Cross-Attention [4] on language modeling score (as measured by bits-per-byte, BPB), while being more efficient in terms of parameter count and computation cost. Then, we define metrics to measure the utility of the retrieved context and use it to guide the training of the retriever. We further show competitive results on downstream tasks of question answering, and demonstrate that the model takes advantage of the retrieved context without memorizing facts within its internal parameters. Finally, we study the implication of context retrieval in terms of retrieval latency, accuracy and computation cost.

To summarize the main contributions of this article:

- Proposed a novel Encoder-Decoder based architecture for incorporating retrieved external context, which decouples context encoding from language model inference.
- Demonstrated the competitive results of the proposed model on both the auto-regressive language modeling task and the open domain question answering task.
- Analyzed model behavior by understanding how context improves language modeling prediction on tokens with different linguistic properties and how the model performs grounded context transfer.
- Discussed computational cost and retrieval efficiency in context augmentation.

## 2    Related Works

Large language models, typically in the form of big neural networks, are trained with a huge amount of training data rich in unstructured knowledge. Researchers have found that after model training, the neural networks often end up storing a surprisingly large amount of memorized information within its weights [2, 7] which are then leveraged as a knowledge base. Multiple hypotheses have been developed on how components such as fully-connected layers [13] and attention layers [5] may be responsible for such memorization behavior. While the capability of storing world knowledge is desirable, memorization also contributes to huge model sizes and the lack of explicit control over knowledge base, such as performing selection or updates.

An alternative strategy is to enable language models to incorporate world knowledge in the form of retrieved context from external sources, instead of having to memorize them. Multiple works have proposed architectures that support external retrieval, usually composed of a context retriever that searches a large external key value store and a method of integrating retrieved information.

There are various ways to construct the key value store. The keys are primarily used for similarity matching, and they can be sparse vectors such as BM25 [34], or dense embeddings extracted from part of the model [21, 25, 40], or from pretrained embedders [12, 28], or embedders trained for specific downstream tasks [15, 20, 41, 35, 24, 11]. The values also have various different forms. For example, TOME [11] stores a dense embedding about the contextual information of an entity mention, while Realm [15], RAG [26], FID [18], Retro [4], MARGE [25], DenSPI [35] and DensePhrases [24] store the raw text as the value. Works such as $k$NN-LM [21] and Spalm [44] store one token as a value. Finally the key value store is searched over using vector similarity techniques, typically with some off-the-shelf nearest neighbor search implementations such as FAISS [19], ScaNN [14], HNSW [27] or SPTAG [9].

There are also many different ways to integrate the retrieved context. A popular approach is to concatenate the retrieval results with the original input and jointly process them. It has been adopted by works such as Realm [15], RAG [26], and FiD [18]. Other works utilize some forms of cross attention for the context integration, such as the Chunked-Cross-Attention with input conditioning in Retro [4], $k$NN Attention in Memorizing Transformer [40], Memory Attention in TOME [11] and Cross-Attention in MARGE [25]. For token level integration, $k$NN-LM [21] uses simple linear interpolation while Spalm [44] uses a learned gate based on the last layer embedding. There are also works that directly utilize the retrieval results without any integration, such as DenSPI [35] and DensePhrases [24]. Most of the works use retrieval as a way to augment tasks such as language modeling or question answering, with the exception of MARGE [25] where retrieval is treated as an autoencoder bottleneck for multilingual pretraining, and is not strictly necessary for inference. We compare representative previous works and contrast with our proposal in Table 1.

| Method | Retrieval Granularity | Retrieval Encoding | Context Integration | Decoupled Context Encoding | Tasks |
|---|---|---|---|---|---|
| $k$NN-LM [21] | Token | Last layer | Interpolation | Yes | LM |
| Spalm [44] | Token | Last layer | Gating | Yes | LM |
| Realm [15], RAG [26], FID [18] | Input | Trained | Concat | No | OpenQA |
| Retro [4] | Chunk | Frozen | Chunked-Cross-Attention | No | LM, OpenQA |
| Proposed | Chunk | Frozen / trained | Encoder-Decoder Cross-Attention | Yes | LM, OpenQA |

Table 1: Architectural differences between previous retrieval augmented model and ours in (i) context retrieval, (ii) context integration and (iii) targeted applications.

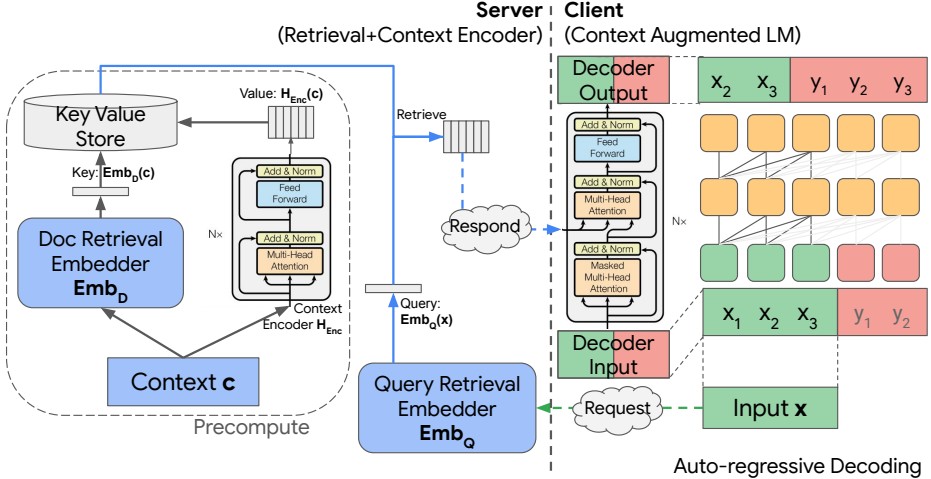

Figure 1: **Architecture of context augmented language model.** We opt to use the standard `Encoder-Decoder` cross attention mechanism for context incorporation which allows us to decouple context encoding from LM inference. $\mathbf{c}$, $\mathbf{x}$, $\mathbf{y}$ serve as context, decoder input and decoder target respectively. In other words, the client sends over input $\mathbf{x}$, and the server conducts retrieval to find relevant context and returns the encoded representation $\mathbf{H}_{\mathtt{Enc}}(\mathbf{c}; \theta_{\mathtt{Enc}})$. The encoded representation $\mathbf{H}_{\mathtt{Enc}}$ is pre-computed offline and returned as "metadata" of the retrieval. Note that training of the `Encoder` and `Decoder` is joint, while they are decoupled at inference time: the client does not need to store parameters or the run inference on the `Encoder` component.

## 3 Architecture

We use `Encoder-Decoder` Transformer architecture [39] to integrate language model input and retrieved context. We denote the context encoder and LM decoder as `Enc` and `Dec` respectively. Given an input token sequence $\mathbf{x} = (x_1, x_2, ..., x_n)$, the task is to predict the next tokens $\mathbf{y} = (y_1, y_2, ..., y_s)$.

Without external context, we concatenate $\mathbf{x}$ before $\mathbf{y}$ and the task becomes a traditional auto-regressive language modeling to predicts targets $\mathbf{y}$ following input (or "promopt") $\mathbf{x}$. In this setting, only the decoder is involved (denoted as "No-retrieval"). To incorporate external context, we use $\mathbf{c}$, $\mathbf{x}$, $\mathbf{y}$ to serve as encoder input, decoder input and decoder target respectively. We first use a retriever to identify the context $\mathbf{c} = (c_1, c_2, \dots, c_m)$ given input $\mathbf{x}$, then fetch the the pre-computed encoder output of the corresponding context tokens $\mathbf{H}_{\mathtt{Enc}}(\mathbf{c}; \theta_{\mathtt{Enc}})$ as output. In this setting, encoder output $\mathbf{H}_{\mathtt{Enc}}$ is directly used by the decoder through Encoder-Decoder cross-attention to influence the final prediction. The decoder does not have to know the exact tokens of $\mathbf{c}$ that are retrieved.

$$\mathtt{No\text{-}retrieval}: P(y_i | y_{<i}, x_1, x_2, \dots x_n; \theta'_{\mathtt{Dec}})$$
$$\mathtt{Retrieval}: P(y_i | y_{<i}, x_1, x_2, \dots x_n, \{\mathbf{H}_{\mathtt{Enc}}(\mathbf{c})\}; \theta_{\mathtt{Dec}})$$

Under this formulation, only decoder parameters $\theta_{\texttt{Dec}}$ are required at inference time. The retriever retrieves indices of the relevant context and looks up their encodings. The context encodings are processed ahead of the time, and are completely decoupled from online operations. This is in contrast to previous works of Realm [15], Rag [26] or Retro [4] where the interaction between input $\mathbf{x}$ and context $\mathbf{c}$ is bi-directional, which necessitates context encoding at inference time. In our model, information flows uni-directionally from $\mathbf{c}$ to $\mathbf{x}$ and $\mathbf{y}$, and that the encoding of each context $\mathbf{c}$ is processed independently. On one hand, this is more restrictive than bi-directional interaction; on the other hand, such a design ensures complete decoupling of context processing and the online language model inference. The exact mechanism is detailed in Figure. 1.

Conceptually the retriever can be an arbitrary blackbox. In practice, we use a dual encoder formulation [8, 16], which first represents $\mathbf{x}$ as a query embedding $Emb_Q(\mathbf{x})$ and performs vector similarity search over a database of $\mathcal{D}$ to find the indices of documents whose document embedding has the highest inner products with the query embedding. We then look up the context encoder outputs that correspond to retrieved indices and return them as the retriever output.

$$l^* = \arg\max_{\mathbf{v} \in \mathcal{D}} \langle Emb_Q(\mathbf{x}), \mathbf{v} \rangle); \mathcal{D} = \{Emb_D(\mathbf{c}); \mathbf{c} \in \mathcal{C}\}$$

$$\mathbf{H}_{\texttt{Enc}}[l^*] = \texttt{Enc}(\mathbf{c}_{l^*}; \theta_{\texttt{Enc}})$$

In the case of multiple supporting context, $k$-$\arg\max$ is used instead of $\arg\max$. The encoder outputs of each supporting context are then concatenated:

$$P(y_i|y_{<i}, x_1, x_2, \ldots x_n, Concat(\mathbf{H}_{\texttt{Enc}}[l_1], \mathbf{H}_{\texttt{Enc}}[l_2], \cdots \mathbf{H}_{\texttt{Enc}}[l_k]); \theta_{\texttt{Dec}});$$

Where $Concat$ is simply vector concatenation:

$$Concat((\mathbf{h}_1, \mathbf{h}_2, \cdots, \mathbf{h}_n], [\mathbf{g}_1, \mathbf{g}_2, ..., \mathbf{g}_m], ...) = [\mathbf{h}_1, \mathbf{h}_2, ..., \mathbf{h}_n, \mathbf{g}_1, \mathbf{g}_2, ..., \mathbf{g}_m, ...]$$

At training time, the encoder and decoder are jointly trained. We first perform offline retrieval to form triplets of $(\mathbf{x}, \mathbf{y}, \mathbf{c})$, where $\mathbf{c}$ is retrieved by some predefined retriever. The loss is masked and only defined on the targets $\mathbf{y}$. Because the encoding of each context is independent and there is no interaction between context, the attention matrix of encoder is block diagonal and we process them in a linear loop over each diagonal block. Thus, the computation cost of both encoder and decoder at each step is linear in the number of context.

For online language model inference, only the retriever and the decoder are involved. The retrieval embedding $Emb_D(\mathbf{c})$ and the encoder output $H_{\texttt{Enc}(c)}$ of the context are both offline pre-processed and prepared into the retrieval database of such associated key-value pairs. When a new input sequence arrives, the retriever is only responsible for the approximate nearest neighbor search and lookup of the associated value that is the pre-computed encoder output. The decoder then takes in the input sequence and cross attends on the concatenation of pre-computed encoder output to generate the targeted tokens. Thanks to the decoupling, neither the retriever nor the decoder needs to store encoder parameters. Hence, such an approach is more parameter efficient compared to similar works such as Retro [4] by saving the storage and computation budget on encoder, which is helpful in "client-server" scenario where the capacity of the "client" can be limited. When accounting the parameter count in comparison with other models, we only need to count the decoder and cross attention parameters. We also followed Retro's [4] approach of excluding the embedding matrices from the parameter count.

## 4 Auto-regression Language Modeling

We experimented with the same "encoder-decoder" context incorporation mechanism for both auto-regressive language modeling and open domain question answering. The only difference is that auto-regressive language modeling processes input sequences in a sliding window fashion, while question answering task receives the full input sequence (the question) at once.

### 4.1 Experimental Setup

For auto-regressive language modeling, we use English C4 [32] version `2.2.1`, the same as Retro. We train the language model and prepare the retrieval database using the `train` split and evaluate the results using `validation` split. The language model target sequence is a sliding window (chunk)

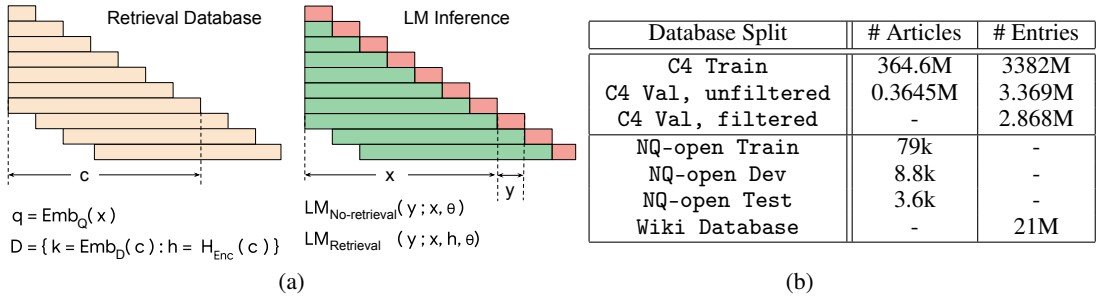

| Database Split | # Articles | # Entries |
|---|---|---|
| `C4 Train` | 364.6M | 3382M |
| `C4 Val, unfiltered` | 0.3645M | 3.369M |
| `C4 Val, filtered` | - | 2.868M |
| `NQ-open Train` | 79k | - |
| `NQ-open Dev` | 8.8k | - |
| `NQ-open Test` | 3.6k | - |
| `Wiki Database` | - | 21M |

(a)  (b)

Figure 2: (a) Chunking scheme for language modeling training with C4. For each article in C4, we divide the tokenized text into non-overlapping blocks of at most 64 tokens and use them as targets $\mathbf{y}$. We use the preceding tokens of at most 448 in length as $\mathbf{x}$. (b) The number of entries in the retrieval database of C4 auto-regressive LM and Natural Question QA task.

of $s = 64$ tokens, with at most $n = 448$ preceding tokens are used as input sequence. This setup is similar to XLNet [43] and Retro [4]. The target and input sequences that are smaller than the given window size (64 and 448, respectively) are padded with zeros.

To construct the retrieval database, the same sliding window processing is also used for the context sequences. The database is formed as associated pairs of retrieval embedding and encoder output: $\{Emb_D(\mathbf{c}) : H_{\texttt{Enc}}(\mathbf{c})\}$, where context $\mathbf{c}$ are the sliding window of 512 tokens with a stride of 64 tokens. We choose our hyper-parameters to be comparable to Retro: chunk size $s = 64$ and input window size $n = 448$ (smaller than 2048 that of Retro). This also implies that the number of entries in the database is larger than the number of articles, but smaller than the number of tokens.

Our training corpus is in the form of triplets $(\mathbf{x}, \mathbf{y}, \mathbf{c})$. $\mathbf{x}$ and $\mathbf{y}$ are acquired directly by applying sliding window on the `train` split of C4. Then BM25 [34] is used as a bootstrapping retriever to mine relevant context $\mathbf{c}$ to from the database. The first retrieval results with no more than 8 consecutive token overlap with the target is used as context. Figure 2a illustrate the sliding window construction of database as well as sequence served as input and targets. Table 2b gives the exact number of entries in the resulting database used as external context.

We use mT5 [42] as the backbone architecture for our context augmented `Encoder-Decoder`, and train our models from scratch. `Train` split is used both for training and retrieval, while `validation` split is used for bits-per-byte (Bpb) evaluation. In auto-regressive language model evaluation, due to the fact that text are crawled from web sources, there can be a non-trivial overlap of tokens between the training and validation splits. In such cases, tokens are often "copy-pasted" from retrieved context into targets without changing. Such copying leads to near-zero perplexity on targets and has a big effect on final bits-per-byte measurement. Following the discussion on dataset leakage of Retro [4], we filtered any example whose targets and context sequences have more than 8 common consecutive tokens (correspond to 12.5% filtering of Retro). We found 14.87% of the validation chunks are removed by the filtering of longest common substring.

Figure 3a reports our auto-regressive language model results with different model sizes under the Bit-per-byte (bpb) values. Bpb is tokenizer agnostic and is often used to compare models with different vocabulary. We use mT5 `base`, `large` and `XL` respectively without modification, and the results compare favorably to Retro models with similar sizes that uses customized chunked-cross-attention which couples context encoding and LM decoding. Our experiments demonstrate that context incorporation can be achieved with simple `Encoder-Decoder` cross attention, with the additional benefit of decoupled encoder processing.

## 4.2 Retriever Training

The goal of context retrieval is to identify the context sequences $\mathbf{c}^*$ that maximize the improvement of some utility function, such as the log-likelihood improvement on target prediction. i.e.

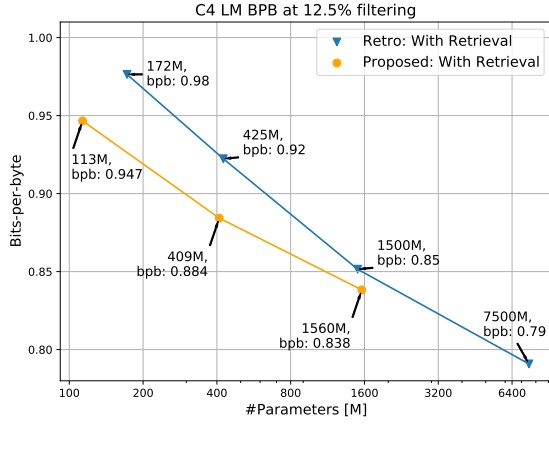

| Model | Model Size | Exact Match Accuracy |
|---|---|---|
| `Realm [15]` | 110M | 40.4 |
| `DPR [20]` | 110M | 41.5 |
| `Ours (Large)` | 409M | 44.35 |
| `RAG [26]` | 400M | 44.5 |
| `Retro [4]` | 7.5B | 45.5 |
| `Ours (XL)` | 1.56B | 47.95 |
| `FiD [18]` | 770M | 51.4 |
| `EMDR [37]` | 440M | 52.5 |
| `FiD + Distill [17]` | 770M | 54.4 |

(a)            (b)

Figure 3: Performance of decoupled encoder-decoder on auto-regressive language modeling (C4) and question answering tasks (Natural Question). (a) Comparing the proposed method and Retro on `c4-en-2.2.1` validation split with 12.5% token overlap filtering. The y-axis measures bits-per-byte (bpb, lower is better), which is the perplexity normalized by token length. The x-axis shows the number of non-embedding parameters in log scale. (b) End-to-end result on Natural Question test split. Following previous works, we measure exact match (EM) accuracy on "short answer type" with at most five tokens.

$$\mathbf{c}^* = \arg\max_{\mathbf{c}} U(\mathbf{x}, \mathbf{y}, \mathbf{c});$$

$$U(\mathbf{x}, \mathbf{y}, \mathbf{c}) = \sum_{y_i} \log P(y_i | y_{<i}, \mathbf{x}, \mathbf{c}, \theta) - \log P(y_i | y_{<i}, \mathbf{x}, \theta')$$

However, it is infeasible to evaluate the utility function $U(\mathbf{x}, \mathbf{y}, \mathbf{c})$ for all triplets $(\mathbf{x}, \mathbf{y}, \mathbf{c})$. Therefore, we introduce a proxy to approximate $U$ by computing the expectation of $U$ conditioned on the token $y_i$ and whether it appeared in the input $\mathbf{x}$ and context sequence $\mathbf{c}$:

$$\hat{U}(\mathbf{x}, \mathbf{y}, \mathbf{c}) = \sum_{y_i \in \mathbf{y}, \mathbf{c}} \mathbb{1}_{y_i \in \mathbf{x}} \bar{U}(y_i | y_i \in \mathbf{x}, y_i \in \mathbf{c}) + \mathbb{1}_{y_i \notin \mathbf{x}} \bar{U}(y_i | y_i \notin \mathbf{x}, y_i \in \mathbf{c})$$

Intuitively, the "context utility" $\hat{U}$ is a weighted token overlap between context $\mathbf{c}$ and target $\mathbf{y}$. The weight is higher if the token $y_i$ did not appear in input but is contained in context, and when the token is "sensitive" to context by showing larger loglikelihood change when the context is present. To train a retriever, we adopted the typical dual encoder formulation with in-batch softmax training similar to that of DPR [20, 16]. We use training data bootstrapped from BM25 retrieval, where (1) the valid retrieval with highest context utility $\hat{U}$ is the positive; (2) the other top retrieval from BM25 that has less than 80% of context utility of the highest one is used as hard negative; (3) The rest of the in-batch samples are regarded as random negatives. All retrievals are subject to the filtering criteria of no more than 8 consecutive tokens.

We trained our retriever using T5X retrieval framework [28] based on mT5-`small` with an embedding dimensionality of 128. The model is trained for 100,000 iterations on a batch size of 8,192 on 64 TPUv3 chips. Unfortunately, using the dense trained retriever does not lead to a visible improvement on the final C4 Bpb evaluation, possibly because both the dense and BM25 retriever end up retrieving similar context. We report results using trained dense retriever because of the better retrieval efficiency.

### 4.3 Ablation and Analysis

To analyze the effect of the context augmentation, we show first in Figure 4a that the context augmentation in language model consistently helps across different model sizes in the proposed

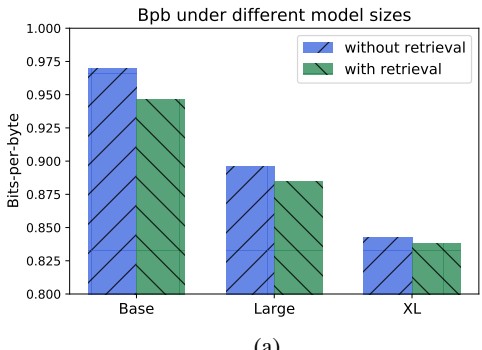
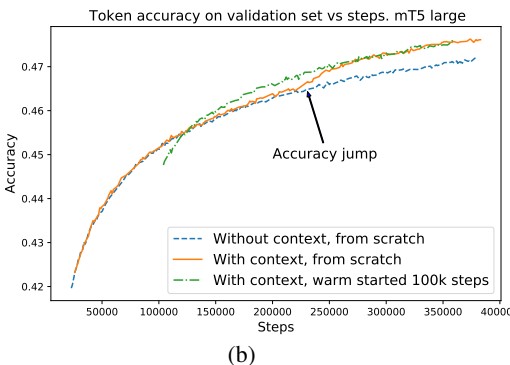

(a)                        (b)

Figure 4: (a) Bpb evaluation (lower is better) comparing models of the same sizes trained from scratch, with and without context retrieval, respectively. (b) Comparing the accuracy history of `Large` model training with and without retrieval. Notice that the curve "jumps" when training with retrieval but without warm-starting.

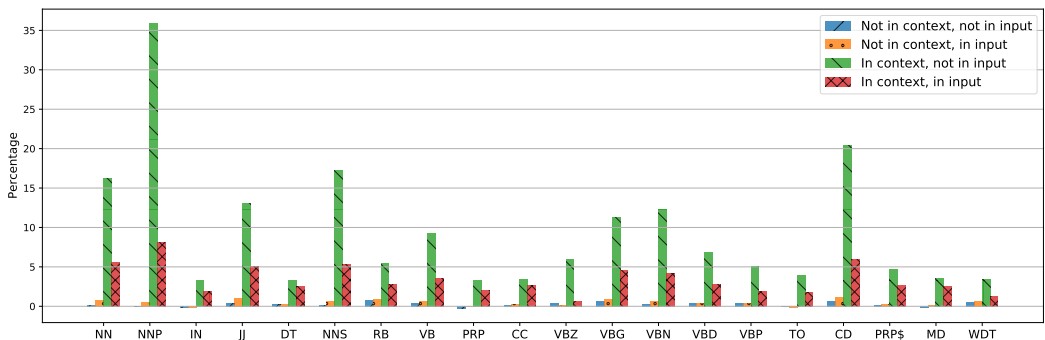

Figure 5: Percentage of improvement on target log-likelihood when there's context. We show the most frequent 20 POS tags, excluding punctuation. For the data on all the POS tags, see Appendix A.

method. While the absolute improvement on Bpb over "No-retrieval" may seem small when it is filtered to 12.5% in longest common sequence, we note that the improvements were made on a limited set of important tokens. Intuitively, tokens of functional words such as "the", "a", "to" should not benefit much from external information and can be predicted with its local context. Yet functional words make up a significant portion in perplexity because of their high frequencies in text.

To this end, we consider the difference in log-likelihood of two language models of same size, trained with and without retrieved context. We evaluate the models on validation split of C4 and obtain a breakdown using part-of-speech (POS) tagging. We used `NLTK` [3] to assign each target token to a POS tag. Note that the tokenization boundaries of `NLTK` and sentencepiece [22] could be different, and the assignment is done by majority voting in that case.

From Figure 5, it is immediately clear that the impact of context retrieval on different types of tokens is not uniform. Noun (`NN`, `NNP`, `NNS`, `NNPS`), and number (`CD`) benefited the most from the context, followed by adjective (`JJ`) and verb (`VB`, `VBZ`, `VBG`, `VBN`, `VBD`, `VBP`). Tokens such as preposition (`IN`), coordinating conjunction (`CC`), etc. are less helped by context. We believe it is because their prediction usually only depends on immediate local context. We also found that whether the target token appeared in the context sequence or input sequence has a large impact on the improvements, and included this in the breakdown. Intuitively, tokens that are grounded in the retrieved context **c** but did not appear in the input **x** benefit the most from context retrieval.

### 4.3.1   Training Details

We trained the Encoder-Decoder LM model for a total of $1,100,000$ steps with a batch size of 512 and a default learning rate schedule of square-root decay. This corresponds to $10,000$ warmup steps with a fixed learning rate of $0.01$, followed by square-root decay for $990,000$ steps. We found training from scratch and fine-tuning on pre-trained models result in similar performance on C4. Therefore

we opted to train from scratch. We also found it beneficial to include additional $100,000$ steps with a lower learning rate of $2 \times 10^{-4}$ after standard training schedule, which slightly improves the results. For `base` and `large` we used 64 TPUv3 chips whereas 128 TPUv3 chips for training XL. XL runs 1.5 training steps per second. XXL is left out due to the insufficient computation resources.

Comparing the training curve of model with and without retrieval, as depicted in Figure 4b, we noticed a phenomenon where both models have roughly the same performance initially but the one with retrieval suddenly increases at some point and continues to improve afterwards. This is especially true in `large` and XL. We suspect that decoders of larger models have higher memorization capacity, and in the initial phase they can improve training objective without delegating memorization to context retrieval. This is a challenge to training stability and one way to address this is by first "warm start" the model with $100,000$ steps of training on a subset of training data with highest 10% of the context utility before proceed on the full dataset.

## 5 Open Domain Question Answering

### 5.1 Natural Question

Large language models are useful because they generalize to downstream tasks, in addition to performing auto-regressive language generation. To demonstrate that the proposed context augmentation scheme is effective on downstream applications we evaluated our proposed model of decoupled `Encoder-Decoder` on the OpenQA task of Natural Question [23].

We use the same question and context processed by [18], where the context is retrieved with DPR retriever [20]. We construct each context sequence in the format of `"title: {title} source: {source}"` and pad or trim it to 256 sentencepiece tokens. The context sequences are then encoded by the encoder independently and simply concatenated, to ensure there is no interaction between them.

The decoder is trained to receive and respond on the sequence containing both the question and answer in format `"question: {question} \n answer: {answer}"` similar to [4], where `"question: {question} \n answer: "` is the input prompt (and there is no loss defined on these tokens) and `{answer}` is the target sequence the decoder is expected to predict. Again, the decoder cross-attends to the concatenated encoder output $\mathbf{H}_{\texttt{Enc}}(\mathbf{c})$ from all the retrieved context and the decoder is completely decoupled from encoder. It does not even have to know the exact tokens of context sequence being returned by the retriever.

The model weights are initialized from T5.1.1 checkpoints [32] We jointly fine-tuned the encoder and decoder for $40,000$ steps with 20 context passages for each input of the `train` split, and validated on the `dev` split every 1,000 steps. We used a batch size of 64, a fixed learning rate of $10^{-4}$ and Adafactor optimizer [36]. Finally, we selected the checkpoint with the best validation accuracy on the `dev` split and evaluated on the `test` split. Each model is trained on 64 TPUv3 chips, and the evaluation metrics is string exact match with SQuAD [33] normalization. We report our final results in Table 3b. Overall, the decoupled `Encoder-Decoder` produced competitive results on Natural Question task, albeit being a much simpler architecture and not specifically designed for QA tasks. We note that FiD [18] uses similar architecture. However, FiD concatenates the question with the retrieved context passages. Therefore the context encoding is dependent on the input sequence and needs to be computed at inference time.

### 5.2 Grounded Answer Generation

Context augmented models tend to generate answers by transferring tokens from context to the output. It is interesting to quantify how often the model output comes from grounded transfer of context tokens and how often this leads to the correct answer. We use the same model trained in 5.1 to (a) run language model decoding with the original set of retrieved context to obtain "original output": $\mathbf{y}^* = Decode(\mathbf{x}, \mathbf{C}; \theta)$; and (b) rerun model prediction but remove all context passages that contain the "original output": $\mathbf{y}' = Decode(\mathbf{x}, \mathbf{C}' = \mathbf{C} \setminus \{\mathbf{c}; \mathbf{y}^* \subseteq \mathbf{c}\}; \theta)$. In the case of grounded context transfer, we expect the "secondshot output" to change, but still grounded in the remaining context.

We show grounded-ness analysis in Table 2. Overall, we found 76% (2,765 out of 3,354) of the cases are likely due to grounded context transfer, because the "secondshot output" has changed to

| Original output $\mathbf{y}^* = Decode(\mathbf{x}, \mathbf{C}; \theta)$ | Original output in context | | | | Original output out of context |
|---|---|---|---|---|---|
| | 3354 | | | | 256 |
| Secondshot output $\mathbf{y}' = Decode(\mathbf{x}, \mathbf{C}'; \theta)$ $\mathbf{C}' = \mathbf{C} \setminus \{\mathbf{c}; \mathbf{y}^* \subseteq \mathbf{c}\}$ | Output changed still in context $\mathbf{y}' \neq \mathbf{y}^*, \mathbf{y}' \subseteq \mathbf{C}'$ | Output changed out of context $\mathbf{y}' \neq \mathbf{y}^*, \mathbf{y}' \not\subseteq \mathbf{C}'$ | Output unchanged | Excluded | - |
| | 2765 | 261 | 291 | 37 | - |

Table 2: Analysis of grounded context transfer on Natural Question. First row shows the number of questions broken down by whether the original model output appeared in the retrieved passages. To demonstrate the model follows grounded context transfer, we removed context passages containing original output and reran inference on the remaining context passages. 76% of output changed due to the removal of "original output" but the new prediction is still contained in remaining context.

something different from "original output", but is still grounded in the remaining context passages. In addition, we found that the accuracy is 51.5% when model output is from grounded transfer, higher than 35.4% for the rest. This implies that the model has higher accuracy when the output is grounded. We exclude 37 examples from the analysis because they have less than 20 neighbors after removing original output.

# 6   Computational Discussion

We evaluate the computation implications of our method around inference latency and space-time trade off. We estimate the latency based on benchmarks and publicly available packages and performance data. In the Natural Question experiment, we use the 1.56B XL model in Table 3b, whose encoder output have dimension of 2048 and the activation data type is `bfloat16` [1]. When profiled on a single TPUv3 core, it takes 200ms to encode all 20 retrieved context passages each containing 256 tokens (with paddings). This is equivalent to 10ms per context. Thanks to the decoupling of the context encoding, the context passages encoding is done offline, and simply looked up at inference time. Inferencing the DPR query embedder model takes around 8ms unbatched on one TPUv2 chip. We use ScaNN [14] to perform approximate similarity search on the DPR embeddings with a dimensionality of 768, which takes roughly 12ms to retrieve 20 neighbors with recall@20=0.97 on a single CPU core. For each passage, the uncompressed encoded embeddings have a total size of 1 MiB (`sizeof(bfloat16) * 2048 * 256`). When reading the encoding from an NVMe SSD with a read throughput of 660 MB/s, the lookup takes around 1.5ms and we use another 0.8ms to transmit of one encoding over network. End to end, it takes around 66ms (`(1.5 + 0.8) * 20 + 12 + 8`) to retrieve 20 encoded passages. This is about a third of time for running inference of the encoder, and everything except query embedder is conducted with generic hardware without accelerators which is cheaper and consumes less energy.

However, one limitation of this approach is the amount of disk space and network bandwidth needed to store and transmit all the pre-computed encodings, which scales with the dimensionality of encoder output. We leave it for future research to reduce the encoder output size while maintaining similar quality, which may be possible through compression, dimensionality reduction, or adding projection layer to the encoder.

# 7   Conclusion

In this paper, we advocate the design of context augmented language model based on `Encoder-Decoder` architecture. `Encoder-Decoder` models are simple, proven and enjoy the unique computational advantage that the context encoding and language generation are decoupled. That is, the context encoding can be produced with offline pre-computation and caching, while the decoder is parameter efficient because it is agnostic to the encoder computation. We further demonstrated the effectiveness of this simple architecture by comparing with competitive baselines on common NLP tasks such as language modeling on C4 and question answering on Natural Questions. We also analyzed the model behavior and showed the context augmentation resulted in big improvements on content words and the model generates output grounded in retrieved context most of the time. Finally, we estimated the latency improvement from caching enabled by the decoupled computation.

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
