# OpenReview forum: "Decoupled Context Processing for Context Augmented Language Modeling"
_NeurIPS.cc/2022/Conference — NeurIPS 2022 Accept_

### Official Review · Reviewer_PRTw · 2022-07-02

**Rating:** 8
**Confidence:** 4
**Soundness:** 4 excellent
**Presentation:** 3 good
**Contribution:** 3 good

**Summary:**

This paper proposes a retrieval-augmented generator/language model, based on an encoder-decoder LM.
Unlike other recent papers in the literature, the focus here is on a model that:
1) doesn't have a complex, custom architecture (unlike, e.g. RETRO), instead using the well-studied and popular seq2seq encoder-decoder transformer architecture.
2) has cheaper inference because context encoding can be done offline, and cached.
The proposed model accomplishes this by sacrificing bidirectionality compared to the quite-closely related FID architecture.
The result is a model that will have much cheaper and faster test-time runtimes than models that must process context on-the-fly.
The model is trained by using the encoder part of the model for contexts only. The input is prefxed to the output in the decoder. The authors propose an interesting context-retrieval training setup, that I do not recall seeing this precise formulation before.
The model also compares favourably to RETRO both in term of language modelling and Open domain QA.
The authors go the extra mile by providing a series of interesting and thoughtful analyses.



**Questions:**

Why mt5 vs other t5s?
Why doesn't initialisation from pretrained models work? - I'd like to see numbers or more detail here.

typos:
L1: context retriever -> a context retriever
L37: metrics -> metrics
L118: extra bracket

**Limitations:**

I think the authors do this well in section 6

**Strengths And Weaknesses:**

Strengths:

I really like this paper.

* the paper is well-written and easy to follow, with sensible notation and equations where needed.
* Identifying the cost of computation of retrieved items as an important problem is a good insight, and the proposed solution is elegant.
* the depth of the analysis is really excellent- I enjoyed reading these sections and learnt a lot.
* the simplicity and elegance of the architecture, compared to e.g. RETRO is inspiring
* I am intrigued by paragraph 220-227, I find this an interesting and simple fix to get the context conditioning to work.
* Improved downstream results compared to RETRO on a "practical" NLP application (NQ),


Weaknesses:
* the figures are quite complicated and bit hard to follow (esp fig 1).
* Not sure I fully buy the argument about the encoder not needing to contribute to the parameter count in - the retriever parameters should also count in my view (by the way)
* Add the parameter counts of other models to Table 3b - some of these models that perform better are significantly smaller than both RETRO and your model.
* Not a big weakness, but it is a shame the dense retriever isnt better than bm25, I would expect it to be better.
* Whilst the model is comparatively strong on NQ c.f. RETRO, I am still a bit disappointed by this number for a 3B param model
* Also,  although the context can be decoupled with your architecture, the encoder is fine-tuned, so it needs a full index refresh over the retrieval corpus for the caching argument to work,  which isn't needed for models like FID (in the limit, this is will be more efficient, but we should be careful to be precise).
* Missing/relevant work to compare to, or at least mention in RW: DensPI/DensePhrases (https://arxiv.org/abs/1906.05807,https://arxiv.org/abs/2012.12624), MARGE (https://arxiv.org/abs/2006.15020), Memorizing Transformers (https://openreview.net/forum?id=TrjbxzRcnf-). In particular, I think the DensePhrases work is quite spiritually related.

---

> ### Author Response · Authors · 2022-08-01
> **Response to reviewer PRTw**
>
> Thanks for a very detailed review and suggestions!
>
> > the figures are quite complicated and bit hard to follow (esp fig 1).
>
> We will simplify the figures in the final version.
>
> > Not sure I fully buy the argument about the encoder not needing to contribute to the parameter count in - the retriever parameters should also count in my view (by the way)
>
> This is a fair point. However, for serving large scale models, parameters that matter are those affecting the inference time. Since the encoder part is precomputed and its parameters are not needed during serving, we didn’t count it. For retriever parameters, while we understand the concern, we are following the way of parameter accounting from Retro and FiD to be consistent.
>
> > Add the parameter counts of other models to Table 3b - some of these models that perform better are significantly smaller than both RETRO and your model.
>
> Acknowledged. We’ll update the final version with the parameter count.
>
> > Not a big weakness, but it is a shame the dense retriever isnt better than bm25, I would expect it to be better.
>
> We were also quite surprised by this fact since it is well established that trained neural retrievers outperform BM25 on cleaner, content rich corpus like Wikipedia. However, C4 is a web corpus which contains noisy text (ads, templated text, etc.). We empirically observed that for some of the more noisy text, the retrieval results of BM25 and neural retriever seem to have a big overlap. It may have contributed to the fact that there were noticeable changes to the final BPB scores. We will look into this by further improving retriever training.
>
> > Whilst the model is comparatively strong on NQ c.f. RETRO, I am still a bit disappointed by this number for a 3B param model
>
> Yes, it is correct that the difference in FiD performance and ours is due to the former using joint encoding of query and context at the run time which is much more expensive than ours. Please take a look at our response to reviewer aAnM.
>
> > Also, although the context can be decoupled with your architecture, the encoder is fine-tuned, so it needs a full index refresh over the retrieval corpus for the caching argument to work, which isn't needed for models like FID (in the limit, this is will be more efficient, but we should be careful to be precise).
>
> We agree this could be an overhead. However, please note that there can be other ways to get around it, such as lazy computation.
>
> > Missing/relevant work to compare to, or at least mention in RW: DensPI/DensePhrases (https://arxiv.org/abs/1906.05807,https://arxiv.org/abs/2012.12624), MARGE (https://arxiv.org/abs/2006.15020), Memorizing Transformers (https://openreview.net/forum?id=TrjbxzRcnf-). In particular, I think the DensePhrases work is quite spiritually related.
>
> Thanks for the additional references! We will add them in the final version.
>
> > Why mt5 vs other t5s?
>
>  T5’s tokenizer lacks non-English tokens and would lead to some tokenization errors. mT5 vocab fixed this by including bigger, more non-English tokens as well as byte-fallback.
>
> > Why doesn't initialisation from pretrained models work? - I'd like to see numbers or more detail here.
>
> We agree that our sentence is a bit confusing. We will make it more precise in the revised version. Initializing from scratch and initializing from a pretrained checkpoint requires roughly the same number of steps to achieve similar level of performance. Therefore we opt to initialize from scratch.
>
> > typos: L1: context retriever -> a context retriever L37: metrics -> metrics L118: extra bracket
>
> We would like to thank the reviewer for a very detailed review. We will fix the typos.

---

### Official Review · Reviewer_DW3N · 2022-07-06

**Rating:** 5
**Confidence:** 4
**Soundness:** 3 good
**Presentation:** 3 good
**Contribution:** 3 good

**Summary:**

This paper presents a method to augment the context based on retrieved relevant information, and keep the additional context representation separated from the history/prompt in an encoder-decoder setup, and thus decoder can be separately trained, and there’s latency benefit since the context and query don’t need to be re-encoded jointly after context is retrieved in inference time.  Authors evaluated their methods on language modeling and open domain question answering tasks and showed the computational benefit of the proposed method.


**Questions:**

Authors provided enough details about computing, which is great.

Additional suggestions:
Qualitative analysis would also help.
The paper doesn’t provide any examples. Showing retrieved context and explaining the relevance of that is useful.


**Limitations:**

The discussion is mostly focused on the computing aspect, which I think is appropriate given the paper's main focus.


**Strengths And Weaknesses:**

The proposed method is intuitive. I feel people have done such evaluations but couldn't find an exact paper with such results.
The paper presents experimental results showing the proposed method has good performance (comparable to other approaches while reducing computational cost).

Authors did a good job and provide enough details to show the advantage of their method, mainly in terms of the computational efficiency.

One weakness of the paper is the novelty. As mentioned above, the proposed method is straightforward, making people feel this is a standard practice to save computational cost.  On the other hand, I'm not aware of similar results in prior work, so it's good to see such quantitative comparisons.

About the two sets of experimental results.
For the LMs, the authors showed perplexity results, and also the ablation study. The POS based analysis is good, providing some insights about the retrieved context and why it's helping the model performance.
However, for the QA task, the analysis is not strong. The results are overall acceptable. But comparing the proposed method to FID and other similar ones, is the performance difference purely because of the encoding during inference (jointly encode the query and retrieved context)?  such understanding is important and very relevant to the proposed method.

---

> ### Author Response · Authors · 2022-08-01
> **Response to reviewer DW3N**
>
> Thanks for the detailed review! We’ll address your comments and questions below.
>
> > One weakness of the paper is the novelty. As mentioned above, the proposed method is straightforward, making people feel this is a standard practice to save computational cost. On the other hand, I'm not aware of similar results in prior work, so it's good to see such quantitative comparisons.
>
> While Encoder-Decoder architecture itself is known to the field for a long time, to our best knowledge, this was the first time it was studied in the context of incorporating retrieved information for language modeling. We think the obviousness in architecture is a good thing, since this leads to simplicity in implementation and computational benefits from decoupled context encoding.
>
> > But comparing the proposed method to FID and other similar ones, is the performance difference purely because of the encoding during inference (jointly encode the query and retrieved context)? such understanding is important and very relevant to the proposed method.
>
> Yes, it is correct that the difference in FiD performance and ours is purely due to the former using joint encoding of query and context at the runtime which is much more expensive than ours. Please see our response to Reviewer aAnM for the details.
>
> > Additional suggestions: Qualitative analysis would also help. The paper doesn’t provide any examples. Showing retrieved context and explaining the relevance of that is useful.
>
> We visualized many context/input/target triplets in the appendix and supplementary material, with annotations on how much improvement each target token received and whether the token appeared in the context/input.

---

### Official Review · Reviewer_aAnM · 2022-07-12

**Rating:** 4
**Confidence:** 4
**Soundness:** 3 good
**Presentation:** 3 good
**Contribution:** 2 fair

**Summary:**

This paper explores how to leverage retrieved context for language modeling by decoupled encoder-decoder models, like Retro (Retrieval-Enhanced Transformer). The model mainly includes two components: 1) a retrieval that retrieves the k most relevant texts of input x via vector-based search; then 2) a transformer-based encoder-decoder model is used to predict the target based on both input x and the retrieved context. Compared with Retro, this integration of input and retrieved context is at decoder instead of encoder. The model is evaluated on auto-regressive language modeling on C4 datasets and Natural Question (a question asking task).

**Questions:**

How the bootstrapped retriever training affects the NQ results? Do you have any ablation?



**Ethics Review Area:**

["I don’t know"]

**Limitations:**

This type of training should help more on knowledge intensive or grounded tasks, and it would be great to see more evaluation on them.

The model may work on zero-/few shot transfer settings, and it would be great to have such experiments as well.

It is hard to understand the point of Figure 5, and how it is related to main story.




**Strengths And Weaknesses:**

Strengths:

The proposed approach is simple, and it seems more parameter efficient than Retro.

It provides extensive experiments especially on different scales.


Weaknesses:

1)Many experiments results are about bits-per-byte on C4 dev. I’m wondering how this metric correlated to end-task performance.

2)On the only one real-world application, NQ, the performance is still far behind FiD which is very similar to the proposed approach in terms of model architectures. Nothing that FiD only has 770m parameters

---

> ### Author Response · Authors · 2022-08-01
> **Response to reviewer aAnM**
>
> Thank you for your review! We address your comments and questions below:
>
> > Many experiments results are about bits-per-byte on C4 dev. I’m wondering how this metric correlated to end-task performance.
>
> Bits-per-byte (or similarly, perplexity) is a commonly used metric for autoregressive language modeling tasks on large corpora. Both Retro and KNN-LM have used bpb/perplexity as their primary evaluation of LM, and we also used it to allow fair comparisons with previous works. Empirically, performance of the LM on downstream tasks is highly correlated with the performance on perplexity.
>
> > On the only one real-world application, NQ, the performance is still far behind FiD which is very similar to the proposed approach in terms of model architectures. Nothing that FiD only has 770m parameters
>
> We acknowledge that there is some gap between our method and FiD on short-QA tasks, which is not too surprising since our context (paragraph) is encoded independent of the question. However, please note that:
>  * The main focus in our work is to explore and compare context incorporation mechanisms in a general LM setting. While we may not be the current SOTA on QA task, we demonstrated that it achieves competitive results compared to strong baselines such as Realm/DPR (more specifically targeting QA task) and Retro (also general LM).
>  * Also our architecture is computationally much cheaper than FiD due to ahead-of-the-time encoding of context paragraphs.
>  * We agree that it will be good to do more experiments in the future to examine how big the gap is (FiD’s strategy, vs our question-independent context encoding).
>
> > How the bootstrapped retriever training affects the NQ results? Do you have any ablation?
>
> While we don’t think the retriever trained for ContextUtility task on C4 without finetuning can directly outperform QA retrievers such as DPR, a similar retriever finetuned on NQ task may be able achieve competitive results. For example, recent work such as https://arxiv.org/pdf/2112.07899.pdf (Table 3) has shown a strong performance on finetuned T5 retriever compared to DPR. There are multiple strategies that improve the performance of QA retrievers (e.g. pretraining [14], hard negative mining data augmentation [19,36]). In this paper, we opted to use the pre-computed DPR retrieval result set to eliminate the difference from retrieval results, thus making our results directly comparable to FiD/Retro/DPR numbers.
>
> > This type of training should help more on knowledge intensive or grounded tasks, and it would be great to see more evaluation on them.
>
> > The model may work on zero-/few shot transfer settings, and it would be great to have such experiments as well.
>
> We agree with the reviewer that the model has shown interesting results on grounded knowledge transfer therefore it’ll be interesting to evaluate on more grounded tasks, and since it’s trained on auto-regressive LM tasks, it also has potential to perform few/zero shots tasks. However, it’s currently beyond the scope of this paper and we’ll leave for the future.
>
> > It is hard to understand the point of Figure 5, and how it is related to main story.
>
> Figure 5 breaks down where the improvements are from, w.r.t. part of speech tagging. This demonstrated that predictions of nouns, proper nouns and numeric values improved significantly, potentially indicating transfer of knowledge. Reviewer DW3N commented: “The POS based analysis is good, providing some insights about the retrieved context and why it's helping the model performance.”

---

### Meta-Review · Area_Chair_MCA8 · 2022-08-26

**Recommendation:** Accept
**Confidence:** Less certain

**Metareview:**

The paper contributes a retrieval-augmented LM that offers some nice features compared to earlier work. First, it is based on a seq2seq architecture instead of a custom one (e.g., RETRO), which is a plus as it is simple, well studied, and the overall approach is quite elegant. Second, the model of the paper also offers cheaper inference compared to many prior models, and context encoding in the paper can be precomputed offline.

While the reviewers agreed on these pros, the main concern during the reviewers’ discussion was in relation to the downstream evaluation. The paper presents evaluation on only one downstream task, which is on the Natural Question (NQ) QA task and results are much worse than FiD. However, as pointed out by the authors, the better performance of FiD isn’t so surprising (as the paper’s model encodes contexts independently of the questions, as this is done offline). By contrast, FID comes with much greater computational costs. Ultimately, we agree with the authors’ argument that the most comparable model is RETRO, and the paper does quite well in comparison. It is nice to see the paper’s model outperforming other strong baselines such as REALM, DPR, and RAG. To mitigate concerns about their model’s relative poor performance relative to FiD, the authors might want to bring up computational efficiency (e.g., concrete running time) earlier in the paper.

**Award:**

No

---

### Decision · Program_Chairs · 2022-09-14

Accept